# Subjective feeling of control during fNIRS-based neurofeedback targeting the DL-PFC is related to neural activation determined with short-channel correction

Ambre Godet[1☯], Yann Serrand[1☯], Alexandra Fortier[1], Brieuc Léger[1], Elise Bannier[2,3], David Val-Laillet[1]*, Nicolas Coquery[1]

**1** INRAE, INSERM, Univ Rennes, CHU Rennes, Nutrition Metabolisms and Cancer, NuMeCan, Rennes, France, **2** Inria, CRNS, Inserm, IRISA UMR 6074, Empenn U1228, Univ Rennes, Rennes, France, **3** CHU Rennes, Radiology Department, Rennes, France

☯ These authors contributed equally to this work.
* david.val-laillet@inrae.fr

**Data Availability Statement:** Raw data are available (https://doi.org/10.57745/XOAXC6).

## Abstract

Neurofeedback (NF) training is a promising preventive and therapeutic approach for brain and behavioral impairments, the dorsolateral prefrontal cortex (DL-PFC) being a relevant region of interest. Functional near-infrared spectroscopy (NIRS) has recently been applied in NF training. However, this approach is highly sensitive to extra-cerebral vascularization, which could bias measurements of cortical activity. Here, we examined the feasibility of a NF training targeting the DL-PFC and its specificity by assessing the impact of physiological confounds on NF success *via* short-channel offline correction under different signal filtering conditions. We also explored whether the individual mental strategies affect the NF success. Thirty volunteers participated in a single 15-trial NF session in which they had to increase the oxy-hemoglobin (HbO2) level of their bilateral DL-PFC. We found that 0.01–0.09 Hz band-pass filtering was more suited than the 0.01–0.2 Hz band-pass filter to highlight brain activation restricted to the NF channels in the DL-PFC. Retaining the 10 out of 15 best trials, we found that 18 participants (60%) managed to control their DL-PFC. This number dropped to 13 (43%) with short-channel correction. Half of the participants reported a positive subjective feeling of control, and the "cheering" strategy appeared to be more effective in men (p<0.05). Our results showed successful DL-PFC fNIRS-NF in a single session and highlighted the value of accounting for extra cortical signals, which can profoundly affect the success and specificity of NF training.

## Introduction

Neurofeedback (NF) is a neurocognitive procedure training aiming to assist people in learning to self-regulate their neural activity in a specific brain region [1]. With the help of neuroimaging techniques (*e.g.* electroencephalography–*EEG*, functional magnetic resonance imaging–*fMRI*, functional near-infrared spectroscopy–*fNIRS*), the activity of a specific brain area can be measured in real-time. During NF, an indicator of this brain activity can be presented to the

**Funding:** The present research was funded by the University of Rennes 1, Fondation de l'Avenir, the Benjamin Delessert Institute, and INRAE. A. Godet received a PhD grant from the University of Rennes 1. The fNIRS device used in this study was funded by CNRS INS2I and FEDER. The funders had no role in study design, data collection and analysis, decision to publish, or preparation of the manuscript.

**Competing interests:** The authors have declared that no competing interests exist.

**Abbreviations:** ADHD, attention deficit hyperactivity disorder; BOLD, blood oxygen level dependen; DL-PFC, dorsolateral prefrontal cortex; EEG, electroencephalography; fMRI, functional magnetic resonance imaging; fNIRS, functional near-infrared spectroscopy; GLM, generalized linear model; HbR, deoxy-hemoglobin; HbO2, oxy-hemoglobin; HRF, hemodynamic response function; ROI, region of interest; SC, short-channels.

participant as a metaphorical representation through a sensory modality (*e.g.* visual, auditory). The application of NF-based training protocols covers a wide range of research areas, from enhancing cognitive abilities in healthy individuals, to improving health conditions and behaviors in the context of neurocognitive disorders. The NIRS-based NF approach is an alternative to EEG and fMRI with great potential for translational research and clinical applications [2, 3]. As an optical brain imaging technique, fNIRS is less expensive and more accessible than fMRI, and less sensitive to movement artefacts compared to EEG, leading to a wider range of application in psychiatry and behavioral research among others [4, 5]. Although fNIRS displays lower temporal resolution as compared to EEG and lower spatial resolution as compared to fMRI (*i. e.* fNIRS acquisition signals are restricted to cortical areas), its use is increasing for NF studies due to its portability and low motion sensitivity. In particular, fNIRS-NF training protocols have been used to improve cognitive functions in elderly individuals [6], in impulsive adults [7], as well as in children and adults diagnosed with attention deficit hyperactivity disorder (ADHD) [8–10]. FNIRS-NF can also be applied to improve motor rehabilitation outcomes in stroke survivors [11]. Beyond specific pathologies, fNIRS-NF has been implemented to improve cognitive abilities (executive functioning), such as working memory [12], emotion regulation [13] and cognitive flexibility [14].

Greater brain activity requires increased oxygen consumption and higher cerebral blood flow. With NIRS, brain activity is measured indirectly by sending continuous light in red and in near infrared wavelengths to assess changes in oxy-hemoglobin (HbO2) and deoxy-hemoglobin (HbR) relative concentrations in human tissues [15, 16]. As compared to electrodes in EEG that acquire electrical brain activity, with fNIRS, optods (i.e. paired light sources and detectors), measure changes in oxygen concentration reflecting neurovascular brain activity. In general, two infrared wavelengths are used, where the emitted light penetrates the scalp to reach cortical brain regions. Absorption can be measured through near detectors, from which we can infer changes in HbO2 and HbR concentrations (haemoglobin having different spectroscopic absorption properties when carrying oxygen or not). That being said, one of the drawbacks of NIRS is related to the measure of unspecific blood flow variations that can originate from systemic physiological changes [2, 15, 17]. Indeed, vascularization of non-cortical human tissues, such as skin and skull, can also be detected *via* NIRS. Beyond tissue vascularization, other physiological processes (*e.g.* cardiac rhythm, respiratory rhythm) can also modify HbO2 concentration. In order to distinguish cortical brain activity from other physiological changes (*e.g.* cardiac rhythm: 1–1.3Hz, variations in arterial blood pressure called Mayer waves: 0.09–0.1Hz, breathing: 0.2–0.5Hz), low band-pass filtering is classically used to remove confounding factors from the NIRS signal [15]. Overall, as seen in Pinti et al. ([2019]), band-pass filtering is often used without consensus on the low-pass frequency value, ranging from 0.09Hz to 0.5Hz. Depending on this low-pass cut-off value, the noise contribution of physiological signal changes can be removed from the data. However, the low-pass filter cut-off also needs to be appropriately defined with regards to the stimulation frequency itself [18]. Consequently, discrepancies remain regarding which filter characteristics should be used in order to investigate brain signal activity in a specific experimental design without removing too much of the brain signals of interest [18, 19]. Indeed, for fNIRS-based NF purposes, in which only intermittent neural activation occurs during a NF trial, the filtering option used for other fNIRS applications, such as for evoked-related activation analysis, might not be appropriate. To cope with the contamination of extra-cerebral tissue signals in particular, the use of short-channels (SC) is encouraged [20, 21]. For better insights on the advantages of short-channels corrections, one can refer to Santosa et. al. [22]. By reducing the distance between the light sources and detectors, the depth of the light pathway is reduced, leading to a measure of extra-cerebral tissues signal only. Thus, the SC signals can be used as regressor for subsequent offline

statistical analysis. The use of this approach in fNIRS-based NF studies is poorly documented [23, 24] and to date, most of fNIRS-NF studies have been conducted without SC. Since the fNIRS-NF setup without SC might be better suited for clinical purposes, it is thus important to investigate the link between online NF signal without SC, experienced by the participants, and the offline SC-corrected signal, which can be more reflective of accurate neuronal activation. Recently, some efforts have been made to unify the practices for fNIRS data acquisition and analysis [19].

Different regions of interest can be targeted in NF training protocols. The dorsolateral pre-frontal cortex (DL-PFC) is a common target to train cognitive processing [7, 9, 10, 12, 13, 25–29], given that increased DL-PFC activity would result in an increased ability for cognitive processes. Some studies have shown that individuals are better at controlling DL-PFC activity in NF training than others [9, 12, 13], but these findings remain scarce and more research is needed to confirm its efficacy. To our knowledge, only one study has used a SC regression during post-processing in a fNIRS-based NF training targeting the DL-PFC (10). Another study used SCs to integrate them during the feedback, but with 1.5-mm distance between the SC [30]. The distance for short channels classically ranges between 0.5 and 1.5 cm [31], but a distance under 10 mm is recommended [32]. Thus, there is a need to further investigate whether and how scalp vascularization may affect the NF success before its implementation in a clinical context.

Beyond the confounding factors related to measurement for fNIRS-NF in particular, another limitation of NF in general is the fact that not all subjects respond or succeed in the task. Indeed, NF success depends on the participants' motivation, involvement in the training, and learning capabilities [33, 34]. These factors can contribute to a range of 15 to 30 percent of individuals who are unable to self-monitor their brain activity at the end of a NF training protocol [1, 35]. These individuals are classically referred to as "non-responders" compared to the "responders" who are successful in the task. Moreover, the degree of consciousness of the NF training paradigm, the instructions provided (explicit or implicit) and the type of mental strategy used (such as cheering, concentration or visualization strategies) to control the region of interest may profoundly affect NF success [36, 37]. These strategies relate to self-encouragement, mental effort (*e.g.* mental to-do list or mental arithmetic) or visualizing a specific situation or object (including memory recall), respectively. In accordance with the Consensus on the Reporting and Experimental Design CRED-nf checklist [38], investigating the mental strategies used by an individual is highly recommended, as they are related to NF learning success and should be reported in NF training protocols.

The purposes of our fNIRS-based NF study targeting the DL-PFC was to (i) assess whether individuals are able to voluntarily and specifically increase their bilateral DL-PFC activity (increased HbO concentrations) across a single NF session, (ii) assess the specificity of cortical DL-PFC activation during NF with offline SC correction and under different signal filtering conditions (0.01–0.09 and 0.01–0.2 Hz), since there is no current consensus on filtering option for NF-based NIRS signal analysis, and (iii) explore the mental strategy verbally reported by the participants in relation to their NF success. As the DL-PFC is related to cognitive control, we expected that cheering or concentration strategies would be related to greater DL-PFC activation. We also expected that a subjective feeling of control during DL-PFC-targeted NF would be related to higher brain activation in the DL-PFC channels.

## Materials and methods

### Ethics statement

This study protocol was approved by the ethics committee of Rennes University-Hospital ("Comité d'éthique du CHU Rennes", n˚21.128) and was conducted in accordance with the French and European General Data Protection Regulation 2016/679 and the Declaration of

Helsinki. Free informed consent was obtained from all participants through written reports. The data used and provided in this paper or as supplemental data were completely anonymized, with no means to infer the identity of participants.

## Participants and study procedure

Thirty adult participants, between 19 and 60 years old (Mean (Standard Deviation) = 38 (12) years old), female/male ratio = 1/1, were recruited. As we aimed to explore our neurofeedback (NF) approach and set-up, we expected that 30 participants should be enough for this validation study. They were asked to perform a single functional near-infrared spectroscopy (fNIRS) NF session consisting of 15 trials in which participants were asked to increase the level of a visual gauge positively correlated with their DL-PFC activity. As recommend by Strehl et al. (2014), no explicit strategy was imposed nor suggested, letting the participants identify by themselves the mental strategy that suited them best [34]. Participants were simply informed that the level of the gauge was a metaphor correlated with the activity of the dorsolateral prefrontal cortex (DL-PFC), a brain region involved in goal-oriented behaviors, cognitive and self-control [5]. This kind of neurofeedback task is classified as *explicit*, as the individuals were aware of the meaning of the gauge signal and knew which brain region was targeted. On the contrary, the provided task instructions are classified as *implicit* as no explicit strategy was provided [36]. See Fig 1 for an overview of the protocol design and scheme of NF sessions.

## Behavioral assessments

In accordance with the CRED-nf checklist [38], the subjective feeling of success and the mental strategies used to control the gauge were gathered during an interview after the NF session. The participants perception of controlling the gauge for the entire session was recorded as a binary "yes"/"no" response *a posteriori*. The participants' best strategy for controlling the gauge was also recorded and then classified by the experimenters according to Kober et al. (2013),which included visualization, cheering, concentration [37] and anger (emotion) as an additional strategy. A complete report of behavioral assessment is presented in **S1 Table**.

## Functional near-infrared spectroscopy (fNIRS) neurofeedback

fNIRS signals were acquired with an 8 LED sources ($\lambda_{1|2}$ = 760|850 nm) / 8 APD detectors NIRScout XP (NIRX, Berlin, Germany) system (sample rate = 7.8125Hz) coupled with the NIRStar software (v15.2, NIRx, Berlin, Germany) and Lab Streaming Layer LSL (San Diego, USA, https://zenodo.org/record/6387090). The oxy-hemoglobin signal, converted to concentration changes (in μM) using the modified Beer-Lambert law by NIRStar from raw signal [39]), is considered to reflect cortical activation variations and was used for real-time feedback calculation with a dedicated software implemented in Python. Psychopy was used for feedback presentation to the participant [40]. As presented in **Fig 1B** and **2A**, sixteen channels were dedicated to record the frontal region, one channel was located on the motor region and one detector was used for short-channel measurement of all channels, *i.e.* superficial signal from the scalp (Fig 2A). For the feedback calculation and presentation, only the four channels covering the DL-PFC bilaterally were considered as previously described [41] (Fig 2A, channels: S1-D1; S1-D2; S7-D5; S7-D7). We used a moving average of 2 sec, and a band-pass filter (Butterworth, order 3, 0.01–0.2 Hz) [18, 19]. The mean value of these four channels was computed and subtracted by the mean signal value of the 5 seconds preceding the NF task to account for signal baseline. The visual feedback consisted in a simple gauge related to the online computed signal. The gauge was made of gradations from yellow (bottom) to red (top), the top half for positive changes in concentration and the bottom half for negative changes. The middle of the

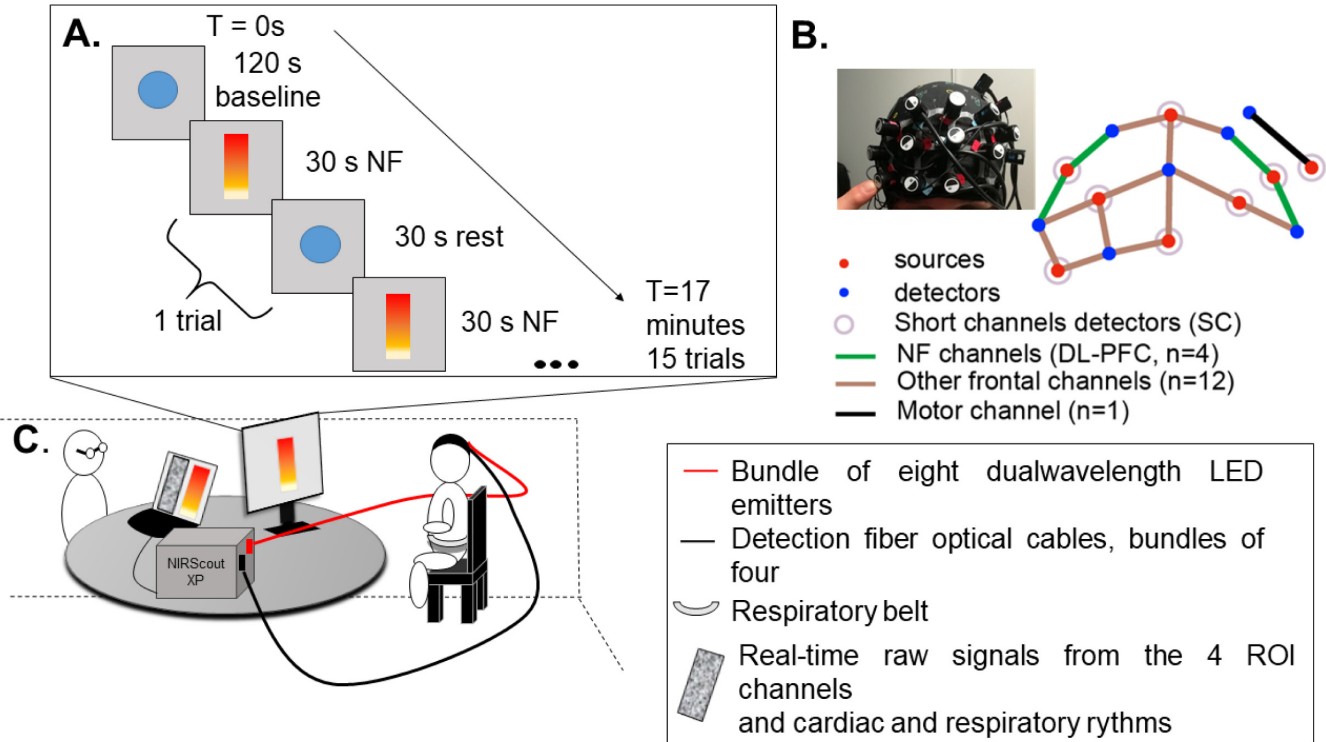

**Fig 1. Experimental design and installation of the participant during real-time neurofeedback training. A.** Protocol design. Each NF session consisted in 15 trials of 30 seconds of NF with the colored gauge, alternating with 15 rest periods with a moving blue circle to help the participants to disengage from the task. During the baseline period, the blue circle also appeared on the computer screen of the participants. **B.** Layout of the NIRS cap with the positions of sources and detectors. **C.** Schematic representation of the installation of the participant in a dedicated dark room (to avoid light noise). The experimenter was hidden behind the participant's computer screen. The NIRScout system was connected to the experimenter's computer to launch the software used for data acquisition and signal feedback. During rest periods, the experimenter could still see the gauge and raw signals, as well as cardiac and breathing rhythms. The task was computed and launched through ©Psychopy and extended on the LCD computer screen (HP Compaq LA2045wg, 1680x1050, 59.94 Hz) placed approximately 80 cm in front of the participant.

gauge consequently corresponded to the baseline recorded just before each NF trial. If there were no changes in concentration, the participant would see the gauge at half height. If oxy-Hb increases, the gauge would increase and if oxy-Hb decreases, the gauge would decrease. According to the Hemodynamic Response Function (HRF), evoked changes in cerebral oxygenation, *i.e.* increased oxy-Hb, reflects increased brain activity [15]. At both ends, the gauge was not linear to avoid "saturation" effects.

The NF session consisted of a two-minute rest, followed by 15 repetitions of: NF task (30 sec) and rest (30 sec). During rest, a concentric blue circle alternatively increasing and decreasing in diameter (0.1 Hz, fixed duration, not related to measured signals) was presented to help participants disengage from the cognitive strategy developed during the NF task. A CRED-nf check list [38] is provided in **S2 Table**.

## fNIRS-based activation analysis

Activation maps were obtained with the NIRS toolbox [42]. The raw data were converted into optical density, then bandpass-filtered (BP filter: Either 0.01Hz to 0.09Hz or 0.01Hz to 0.2Hz) to remove possible confounding signal such as heartbeat, breathing and Mayer waves. Given that no consensus currently exists, we chose to investigate two signal filtering options because a too low low-pass filter might remove too much relevant neural signal while a too high low-

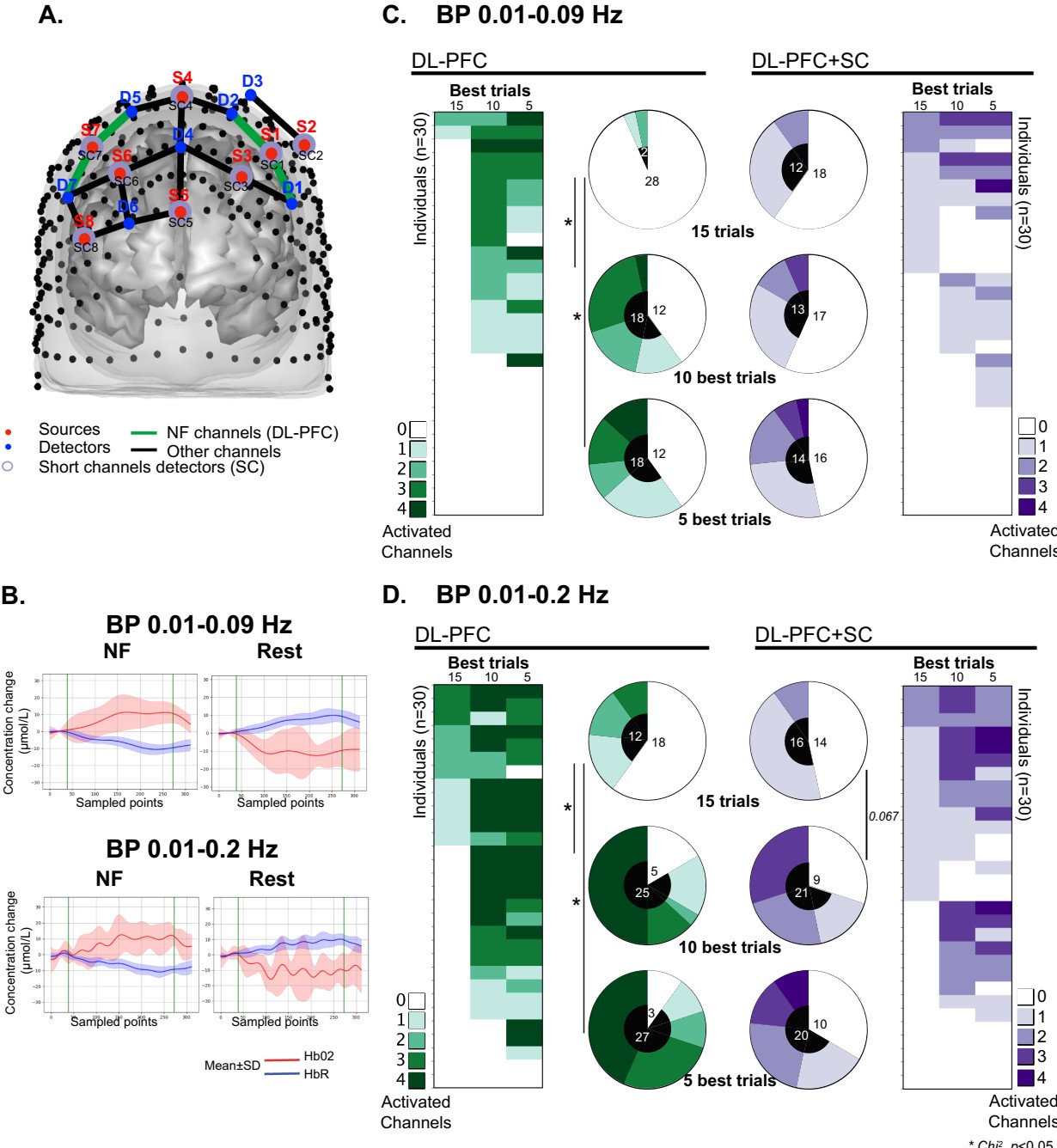

**Fig 2. Number of activated channels as a function of the best-selected trials of the 30 participants (all 15, or only 10 or 5) and two different BP filters. A.** Frontal view of the functional near-infrared spectroscopy (fNIRS) layout with the region of interest (ROI) neurofeedback (NF) channels (n = 4) used for feedback calculation covering the dorsolateral prefrontal cortex (DL-PFC, green). The layout consisted of a coupling of eight sources and seven detectors; Detector 8 was used to collect the short-channel (**SC**) signal for all sources (purple circle). The respective locations of sources (red) and detectors (blue), as well as the labels of sources, detectors and SC detectors are indicated. **B**. Block-average on activated channels within the NF channels from an individual who succeeded in NF training. Red lines represent mean HbO2 concentration and blue lines represent mean HbR concentration. See S1 Fig for the block-average on the 30 individuals for the average signal in the four DL-PFC NF channels for the 10 best trials. Distribution of the participants according to the number of activated channels within the four NF channels for the entire 15 trials, or the 10 and 5 best trials, respectively for **C**: BP 0.01–0.09Hz filter and **D**: BP 0.01–0.2Hz filter. Chi-square comparison of the proportion of participants with at least one activated channel for the 10 and 5 best trials *vs.* 15-all trials, * p<0.05.

pass filter might conversely keep too much irrelevant physiological confounding non-brain signal. The signal was then resampled (native Matlab resample function) at a rate of 4Hz before being converted into oxy- and deoxy-hemoglobin by the modified Beer-Lambert law [39]. A generalized linear model (GLM) based on auto-regressive iterative reweighted least squares analysis was used to model the NF task-related hemodynamic response at the individual level [43]. NF task and rest were added to the GLM model and the NF task onsets were used for activation analysis. This model enables *via* pre-whitening filters to correct serial correlations and outliers caused by motion artifacts. HRF was modeled using the default canonical one from the NIRS toolbox (peak time = 4s, under shoot time = 16s, duration = 32s, ratio = 1/6). Short channel signals were added to the GLM model and eventually used as regressors in order to remove physiological oxy-hemoglobin signal variations from the scalp, which can result from a type of biofeedback (*e.g.* voluntary or involuntary control of heart rate). The standard distance between sources and detectors was between 30 to 42 mm for a long channel and 8 mm for a short-channel, which is the optimal distance in order to detect only non-cerebral NIRS activation for the frontal lobe [17]. For each channel and for each of the 15 trials, beta and *t*-values were computed. Three regions of interest were analyzed: (i) DL-PFC NF region (4 channels), (ii) other frontal brain regions (12 channels), and (iii) a motor brain region (1 channel). The frontal regions other than DL-PFC and the motor region were considered as controls. Based on the number of channels in each studied brain region, a Bonferroni correction for multiple comparison was applied separately for each region of interest. For each participant and brain region of interest, we then computed the number of activated channels as defined by a positive estimator beta-value reaching a Bonferroni corrected statistical threshold of $p_{Bonferroni} < 0.05$. Given that within a single NF session, each participant was possibly exploring different strategies, not all NF trials were expected to yield successful increased DL-PFC activity. We thus decided to investigate the individual channel response in relation with the entire 15 trials, but also the 10 and 5 best trials, determined as the best averaged t-value in the four NF channels computed for each trial (that is to say a higher DL-PFC activation).

## Statistical analysis

Chi-square tests were used with SPSS software (SPSS, Chicago, Illinois, United States) for group comparisons as detailed in the figures' legends (Figs 2–4). A p-value < 0.05 was considered significant. A person blind to the NF data acquisition sessions information performed statistical analyses. As an exploratory approach, we also investigated the effect of gender in some analyses as seen in the figures' details (Fig 4).

## Results

### DL-PFC-targeted NF promotes effective DL-PFC

We performed a GLM statistical analysis on each of the four DL-PFC channels (S1-D1; S1-D2; S7-D5; S7-D7) used for feedback calculation and presentation during the 15 NF trials (Fig 2A, in green). Although the results are based on the GLM statistical analysis, an example of a block-average is depicted in Fig 2B. According to the blood oxygen level dependent (BOLD) HRF, we expected an increase in HbO2 and a decrease in HbR during the NF task, reflecting increased brain activity due to the metabolic demand. In contrast, we expected no relative changes during rest periods, because we did not expect any specific involvement of the DL-PFC in this condition (Fig 2B). Note that the two BP filters promoted similar block average results, even if BP 0.01–0.2Hz showed a more oscillating signal. With the BP 0.01–0.09Hz filtering, when performing the analysis on all 15 trials, two participants had a statistically significant increased activity in at least one NF channel without SC correction (DL-PFC) and twelve

with SC correction (DL-PFC+SC, Fig 2C). Since NF requires training and the participants can explore various strategies to increase the gauge, we cannot systematically expect all 15 NF trials to be successful. Consequently, we also performed an analysis on a selection of the 10 and 5 best trials corresponding to the average of the 10 and 5 best t-value over the four NF channels within the 15 trials for each participant. When considering the 10 best trials, eighteen (60%) participants had increased activity in at least one NF channel without SC correction. For the 5 best trials, no significant difference in the number of participants with at least one activated channel without SC correction was identified (N = 18). With the BP 0.01–0.2Hz filtering, when performing the analysis on all 15 trials, twelve participants had significantly increased activity in at least one NF channel without SC correction (DL-PFC) and sixteen participants with SC correction (DL-PFC+SC, Fig 2D). When considering only the 10 best trials, the number of participants that had an increased activity in at least one NF channel significantly increased to 25 (83%) without SC correction. The selection of the 5 best trials did not result in an improvement in the number of participants with at least one activated channel without SC correction (N = 27). For further analysis, we investigated the channel activation for the 10 best NF trials at the individual level. A table with all channel statistic is available at the DOI link (see data availability section).

## Effect of BP filtering and physiological noise on channels activation

To remove scalp generated physiological noise, we performed additional GLM analyses with the short-channel (SC) signal as regressor for each channel (**Fig 3A**, DL-PFC+SC).

For the BP 0.01–0.09Hz filter, 13 out of 30 participants (43%) were identified as having at least one out of four activated NF channels with the DL-PFC+SC analysis (Fig 3B). Differences were found between the DL-PFC+SC and the DL-PFC analyses (Fig 3B). Three participants who did not present at least one activated channel with the DL-PFC analysis presented an activated channel with the DL-PFC+SC analysis (Fig 3C). Conversely, eight participants with at least one activated channel with the DL-PFC analysis were not detected with the DL-PFC+SC analysis. Overall, both analyses concurred with the detection of 10 participants among 30 (33%) for whom at least one channel was activated within the four channels used for NF. With the SC regression, the activation of at least one out of four NF channels did not reveal any relevant change in the pattern of channel activation in the other frontal brain regions (**Other frontal+SC,** Fig 3B and 3C) nor in the motor brain area (**Motor+SC,** Fig 3B and 3C).

For the BP 0.01–0.2Hz filter overall, 21 out of 30 participants(70%) were identified as having at least one out of four activated NF channels with the DL-PFC+SC analysis (Fig 3B). The DL-PFC+SC and the DL-PFC analyses showed some differences (Fig 3B). Three participants who did not present at least one activated channel with the DL-PFC analysis presented an activated channel with DL-PFC+SC analysis (Fig 3C). Conversely, seven participants who were identified as having at least one activated channel with the DL-PFC analysis were not detected with the DL-PFC+SC analysis. Overall, both analyses concurred with the detection of 18 participants among 30 (60%) for whom at least one channel was activated within the four channels used for NF. With the SC regression, the activation in at least one out of four NF channels was associated with the activation in at least one of the twelve other frontal channels (**Other frontal+SC,** Fig 3B and 3C), and the motor channel (**Motor+SC,** Fig 3B and 3C). This suggests that the detection of DL-PFC activation during NF with the BP 0.01–0.2Hz filter might be less restricted to NF-induced DL-PFC activation compared with the BP 0.01–0.09Hz filter.

As we found only a selective activation of channels in the DL-PFC without an association with the activation in other frontal and motor channels for the BP 0.01–0.09Hz filter, we further analyzed participants' subjective feeling of controlling the gauge with this BP filter.

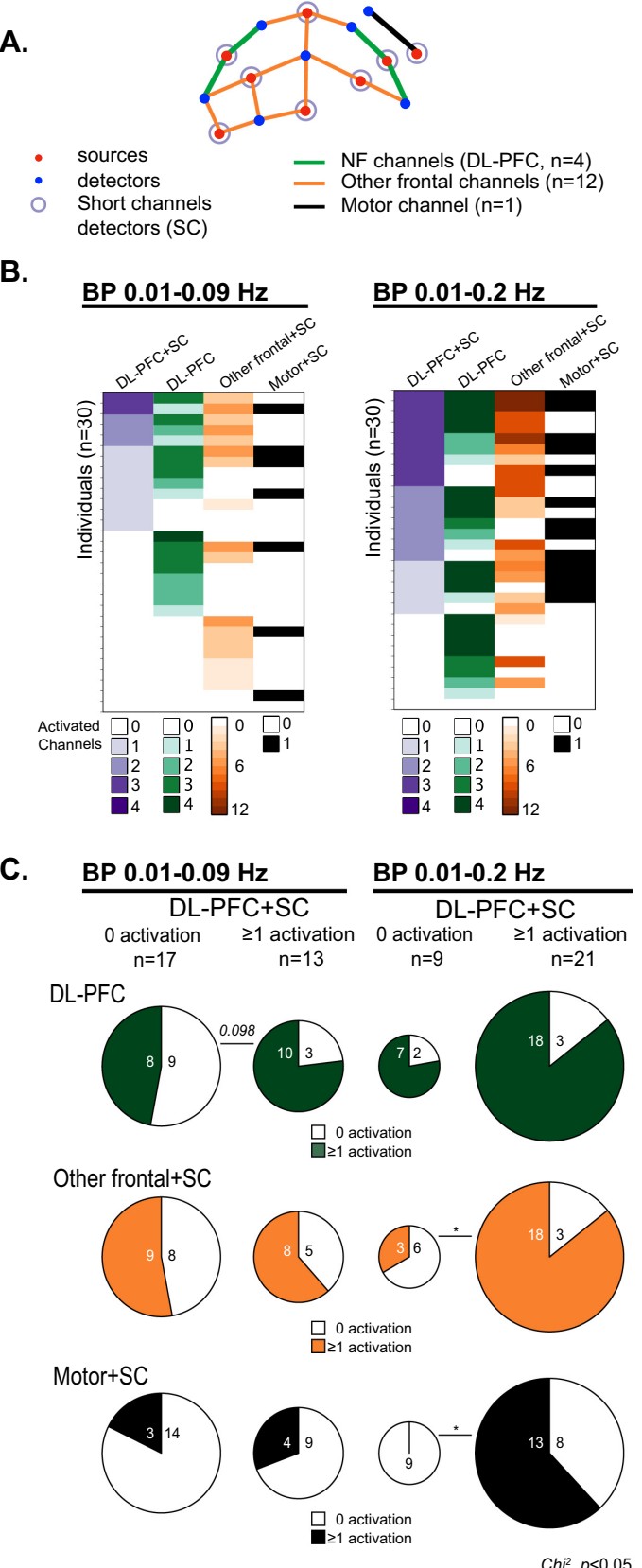

**Fig 3. Characterization of the neurofeedback (NF) signal. A**. Frontal view of the functional near-infrared spectroscopy (fNIRS) layout with (i) the NF channels (n = 4) used for feedback presentation covering the DL-PFC (green), (ii) other frontal channels (n = 12) covering the frontal area (brown), and (iii) one channel covering the motor area in the temporal lobe (black). The layout consisted in the coupling of eight sources and seven detectors; Detector 8 was used to collect the short-channel (**SC**) signal for all sources (purple circle). The respective locations of sources (red) and detectors (blue) are indicated. **B**. For each BP filtering and for each volunteer, the number of activated channels within the four NF channels with (**DL-PFC+SC**) and without (**DL-PFC**) SC regression (*i.e.* exclusion or inclusion of the scalp physiological signal, respectively), as well as the number of activated channels in other frontal (**Other frontal +SC**) and motor areas (**Motor+SC**) with SC regression. **C**. For each BP filtering, influence of the scalp physiological signal on the NF channels activation (comparison of subjects identified as having 0 or at least ≥ 1 activated channel between the DL-PFC+SC and the DL-PFC analyses, *i.e.* with and without SC regression). Proportions of subjects with activation in other frontal regions and motor area in those categorized as having 0 or at least 1 activated channel with the DL-PFC+SC analysis (with SC regression for all). The numbers of subjects for each condition are indicated in the pie charts. Chi-square test for the proportion of the participants with activated channel in function of DL-PFC+SC activation or not, * $p < 0.05$.

## NF feeling of control is related with higher NF channels activation without preferential mental strategy

Among the thirty participants, half reported a positive control feeling while the other half did not experience the feeling of controlling the gauge (Fig 4A). With the DL-PFC+SC analysis of activated channels, we detected a non-significant trend toward a higher number of participants with at least one activated channel within the NF channels when participants reported a feeling of control (Control *vs*. No Control, p = 0.065). Among those who reported a feeling of control, we detected at least one activated NF channel in 9 participants (60%). For the participants that did not report a feeling of control, only four participants (27%) presented an activated NF channel. Interestingly, the difference in activated NF channels related to the feeling of control was more pronounced in women than in men ($p < 0.05$), *i.e.* men reported feeling less control although an activation was observed and all women who showed a DL-PFC+SC activation reported a control feeling. Activation in the Motor+SC channel was more pronounced in the group of participants with no control feeling, whereas activation in Other frontal+SC channels was equally distributed between Control and No Control groups.

We did not identify a mental strategy associated with a better gauge control feeling (Fig 4B) although some strategies seemed to be less efficient to that purpose, such as "Anger" and to a less extent "Concentration". However, we observed that the "cheering" strategy was more associated with a control feeling in men only. The different reported strategies were not associated with DL-PFC+SC activation, and the distribution of the mental strategies used was similar with or without activation (Fig 4B).

## Discussion

In the present study, 30 individuals performed a single session of fNIRS-based neurofeedback (NF) targeting the DL-PFC bilaterally that consisted in 15 trials of 30-sec NF periods alternating with 30-sec resting periods. The aims of this study were to investigate different band-pass filters, accounting for confounding extra-physiological noise, and investigate mental strategies related to DL-PFC activity for the methodological improvement of further NF implementations and applications. For the BP 0.01–0.09 Hz filter without SC regression among the 10 best out of 15 trials, 18 out of 30 participants (60%) had at least one channel of interest significantly activated during NF. This result is in line with the literature, where 15 to 30% of participants are categorized as non-responders to NF [1, 35]. In accordance with Munoz-Moldes' taxonomy, our NF paradigm involved an "*active overt uncued task*" [36]. Indeed, the rewarding element was explicitly presented on a computer screen and participants were aware of its meaning (the gauge level was directly linked to the degree of their bilateral DL-PFC activity)

## A. Control feeling (BP 0.01-0.09Hz)

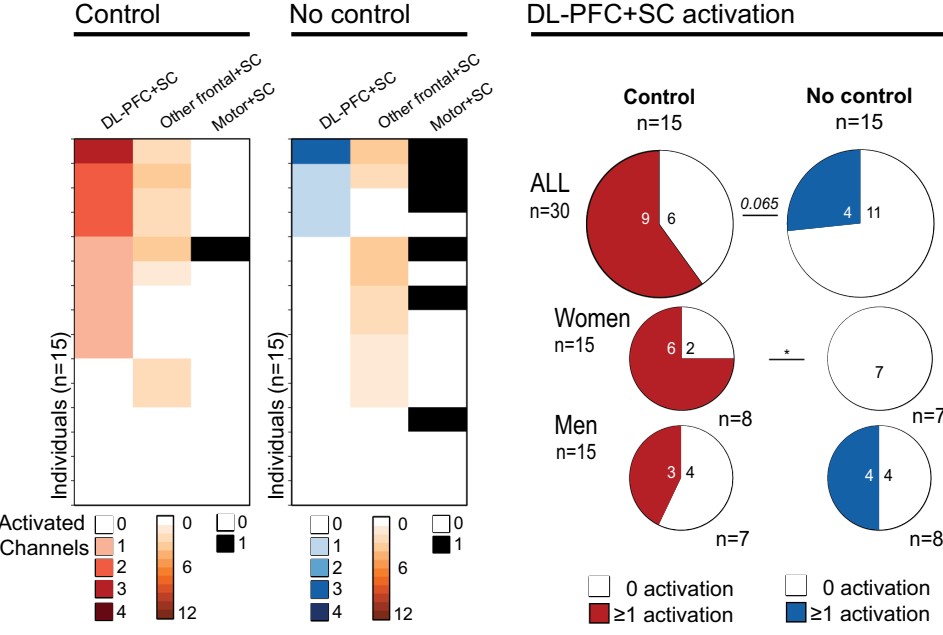

## B. Mental strategy

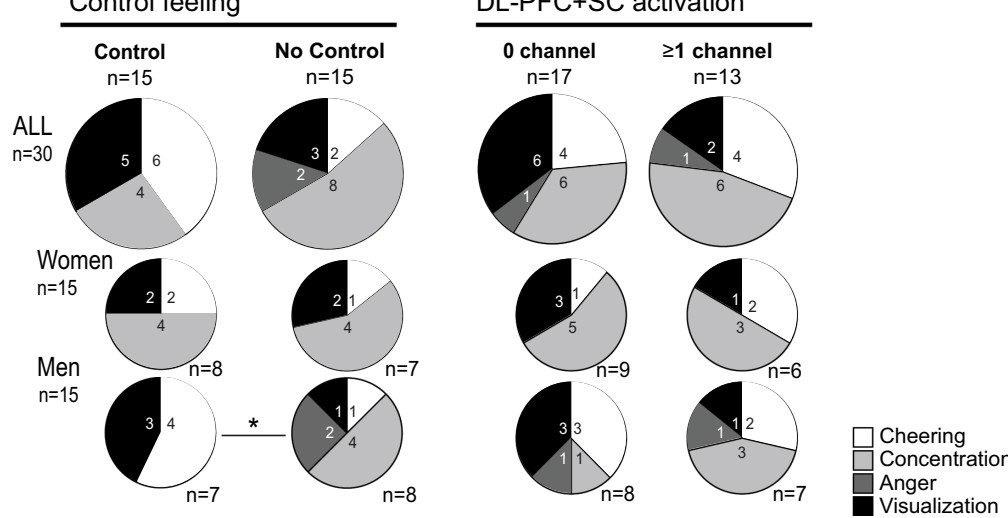

* Control vs No control, Chi², *p*<0.05

**Fig 4.** NF channel activation based on the DL-PFC+SC analysis with the BP 0.01–0.09 Hz filter in relation with **A**. the gauge control feeling (Control vs. No Control), and **B**. the strategy used by the participants to control the gauge. Cheering strategies (*e.g.* self-encouragement); Concentration (*e.g.* being focused on something); Visualization strategies, (*e.g.* visual imagination); Anger (*e.g.* getting angry towards the gauge or himself/herself). Participants who had a positive gauge control feeling are in red while participants with no control feeling are in blue. The number of subjects for each condition is indicated in the pie charts. Chi-square comparison control *vs.* no control, *p*<0.05.

but no explicit strategy was advised [36]. Based on the operant conditioning theory [1, 34], individuals controlled the gauge *via* a series of trials and errors using their own mental strategies in order to find the best one. This may explain inconsistencies across NF trials as some participants tried different mental strategies all along the session.

We found that the BP 0.01–0.09 Hz filter, as previously suggested [18], was better adapted than the BP 0.01–0.2 Hz filter to specifically measure brain activation restricted to NF channels across the DL-PFC. Given that NF might promote intermittent neuronal activation, depending on the participant's success, a wider high-pass filter might not be well-suited when neural activation is not evoked. However, current recommendations suggest using a low-pass filter of 0.5 Hz, as it will not remove too much signal related to brain activation [19]. When considering short-channel (SC) regression, we found that 13 individuals significantly activated at least one channel among four covering the DL-PFC. However, these results also highlight that 8 individuals presented a false positive signal, where non-cortical increased HbO2 contributed to increased NF signal, whereas three individuals presented a false negative signal, as SC regression revealed that at least one channel of interest was activated. These results suggest the importance of accounting for extra-cerebral physiological signals that could bias the NF signal retransmitted to the participants. Indeed, some individuals may have been reinforced towards a mental strategy that was based upon non-cerebral signal. Conversely, others may have abandoned a potentially good strategy of which the effects might have been affected by non-cerebral signals. To address this issue, it could be useful to implement real-time SC regression during NF. An anti-correlation method, based on the assumption that changes in HbO2 and HbR are anti-correlated, could be used as a surrogate approach. However, since HbO2 and HbR are not always anti-correlated, a regression based on this approach could lead to errors [44].

With SC regression and within the 13 individuals for whom we detected at least one activated NF channel, 8 presented an increased activity in other frontal regions and 4 in the motor region. As we provided non-directive instructions and only suggested to engage mental strategies related to cognitive control, some mental strategies used by the participants may have recruited brain areas beyond the DL-PFC. For instance, with SC regression, 4 participants out of 13 showed an activation within the motor cortex. This brain area may have been recruited in relation with the type of mental strategy employed, such as visualization leading to movement picturing. However, the group of participants who declared a subjective feeling of controlling the gauge demonstrated no activated channels in the motor cortex.

Interestingly, we detected activation in the 12 channels covering the frontal brain region, independently of a detected activation in the NF channels. In a fMRI-based NF meta-analysis of healthy individuals, it has been shown that NF training involves the recruitment of many other brain regions beyond the targeted brain region or networks [45]. These regions included the anterior insula, basal ganglia, posterior anterior cingulate cortex, ventro-lateral prefrontal cortex, DL-PFC, premotor cortex, a large cluster in parieto-temporal region and temporo-occipital junction. Thus, NF, independently of the target brain structure, requires or concomitantly induces the activation of brain regions involved in self-regulation and cognitive control, as well as motor and visual regions. This supports the idea that neural mechanisms underlying learning requires cognitive regulation and control, which in turn might explain the widespread PFC activation of some participants in our study.

In addition, a general model of biofeedback learning has been proposed, which involves the generalization of skill learning to higher level cognitive functioning and can be applied to NF and defined below [33]. Thus, participants perform tasks, which elicits brain networks involved in executive functioning, self-control and error detection. Learning also requires the recruitment of brain areas involved in working memory. If the NF paradigm involves a motivating condition (essentially related to voluntary and explicit biofeedback, *i.e.*, the participant

wants to succeed in exerting control over the variable), then the reward system is activated. Finally, learning results in the formation of integrated neural rearrangements through the strengthening of synapses in regions involved in memory [33]. Once learning is integrated, there will be less cognitive involvement in coordinating the neural rearrangements [33]. In line with our result, Gaume's theory and Emmert's meta-analysis described above support the value of repeated sessions to let individuals identify the most effective mental strategies and to consolidate learning processes. Future work should consider examining whether repeated DL-PFC-targeted NF sessions lead to a decrease activation in other frontal brain regions, that could reflect a learning success.

The association between the subjective feeling of control of the visual gauge and its elevation is our objective criterion of success (*i.e.* one channel of interest activated with the DL-PFC +SC analysis). We found that 9 out of 15 participants who reported a feeling of controlling the gauge were right in their self-evaluation. On the contrary, 4 out of 15 participants in the group of participants with no control feeling had at least one channel of interest activated. This suggests that participants experiencing the positive feeling of controlling the gauge tended to better evaluate their ability to perform the NF task compared to others. In addition, when considering the gender of the participants, it appeared that women were better in their subjective perceptions than men. Six upon 8 women had at least one activated NF channel and had a control feeling, whereas nearly half of the men reported no control while there was an activation. Individuals had a relatively accurate perception of control relative to NF channels activation, especially when considering that only a single session of NF was performed with no basal reference. Concerning the strategy used, *cheering* appeared to be more effective in men, although no significant difference was observed between men and women. This result is in line with a previous study that assessed through a single EEG-based NF session an upregulation of the sensorimotor rhythm, where no preferential strategy for efficient NF was reported [46]. In our study, the proportion of mental strategies within groups was not indicative of NF success. Indeed, a single NF session may have led the participant to explore different strategy approaches rather than reinforce a single effective strategy. In addition, some of our participants tried 2 or 3 different strategies (even in the same category), which may be counterproductive and lead to overloaded cognitive resources [37]. In a fNIRS-based NF training protocol targeting the PFC [47], through repeated sessions, the applied strategy "mental to-do-list" was associated with increased activity but only in 5 out of 13 participants. There is consequently no consensus in the literature regarding a preferential effective strategy related to NF success, reinforcing the idea that letting individuals find their personal best strategy is preferable.

## Conclusions

Here, we were able to demonstrate that a single session of 15 repeated NF trials targeting the DL-PFC promoted a significant activation in at least one out of four NF channels covering the DL-PFC for 13 participants among 30. This was after considering the 10 best NF trials and accounting for physiological noise originating from the scalp. The current study should be considered in the light of its strengths and weaknesses. As we did not provide explicit instruction related to a specific mental strategy, 15 trials may have limited the number of possible strategies explored by the participants. More trials or sessions may be required to better identify and reproduce an effective strategy. Moreover, adding a control group in a double-blind paradigm would increase statistical robustness and avoid experimental biases [48, 49]. Indeed, NF training without a control condition contributes to the inability to establish causality, such as to demonstrate the neurophysiological specificity of the NF training, to exclude placebo,

global (spatially non-specific) and behavioral effects [49]. Nevertheless, this kind of experimental paradigm is relevant in early stages of methodological developments (before clinical trial, *e.g.* pilot studies), especially to investigate whether individuals are able to control the region of interest or not. Beyond these drawbacks, we point out that the implementation of SC regression for real-time NF signal computation might be well suited to avoid non-cerebral signal influence. This might be important when the NF mental strategy is investigated.

## Supporting information

**S1 Fig.** Block average of the four NF channels in the DL-PFC, for NF and rest periods and for each individual without short channels (SC) regression and with A. BP 0.01–0.09Hz filter and B. BP 0.01–0.2Hz filter. Individuals are separated in two groups, depending on their feeling of control of the gauge (n = 15 for each group). Individuals who had at least one channel of interest significantly activated during NF are surrounded by a full rectangle, whereas individuals who had no channel of interest significantly activated are surrounded by a dotted rectangle. (EPS)

**S1 Table. Mental strategies.** The mental strategies were acquired by verbal report (interview) at the end of the neurofeedback session in front of two experimenters. These reports were anonymized and then the strategies were classified by the experimenters and approved by another experimenter who was not present during the neurofeedback sessions. Strategies reported as "Cheering" are related to self-encouragement and/or gauge encouragement. "Visualization" strategies are related to visualizing a specific situation or object (including memory recall). "Concentration" strategies are related to any strategy that requires specific mental effort beyond visualization or memory recall. "Emotion" strategies are related to any strategy that was specifically focused on an internal feeling. Because some participants tried different strategies, the strategy that was applied more frequently and/or related to the participants' internal sense of control over the gauge was selected for analysis. Indeed, when a strategy appeared to be more effective for controlling the gauge, it was the strategy used by the participant most of the time. (DOCX)

**S2 Table. CREDlist\*.** Darker shaded boxes represent Essential checklist items; lightly shaded boxes represent Encouraged checklist items. We recommend using this checklist in conjunction with the standardized CRED-nf online tool (rtfin.org/CREDnf) and the CRED-nf article, which explains the motivation behind this checklist and provides details regarding many of the checklist items. (DOCX)

## Acknowledgments

We thank Eugenio Valente (Bionic France) for his technical support.

## Author Contributions

**Conceptualization:** Elise Bannier, David Val-Laillet, Nicolas Coquery.

**Data curation:** Ambre Godet, Yann Serrand, Elise Bannier, Nicolas Coquery.

**Formal analysis:** Ambre Godet, Yann Serrand, Alexandra Fortier, Brieuc Léger, Nicolas Coquery.

**Funding acquisition:** David Val-Laillet, Nicolas Coquery.

**Investigation:** Ambre Godet, Yann Serrand, Alexandra Fortier, Brieuc Léger, David Val-Laillet, Nicolas Coquery.

**Methodology:** Yann Serrand, Brieuc Léger, Elise Bannier, David Val-Laillet, Nicolas Coquery.

**Project administration:** David Val-Laillet, Nicolas Coquery.

**Resources:** Elise Bannier, David Val-Laillet.

**Software:** Yann Serrand.

**Supervision:** David Val-Laillet, Nicolas Coquery.

**Validation:** Yann Serrand, Elise Bannier, David Val-Laillet, Nicolas Coquery.

**Visualization:** Alexandra Fortier, David Val-Laillet, Nicolas Coquery.

**Writing – original draft:** Ambre Godet, David Val-Laillet, Nicolas Coquery.

**Writing – review & editing:** Ambre Godet, Yann Serrand, Alexandra Fortier, Brieuc Léger, Elise Bannier, David Val-Laillet, Nicolas Coquery.

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
