## [Decision Letter · Decision Letter 0]

19 May 2023

PONE-D-22-34461Subjective feeling of control during fNIRS-based neurofeedback targeting the DL-PFC is related to neural activation determined with short-channel correctionPLOS ONE

Dear Dr. Val-Laillet,

Thank you for submitting your manuscript to PLOS ONE. After careful consideration, we feel that it has merit but does not fully meet PLOS ONE’s publication criteria as it currently stands. Therefore, we invite you to submit a revised version of the manuscript that addresses the points raised during the review process.

We look forward to receiving your revised manuscript.

Kind regards,

Xiong Jiang

Academic Editor

PLOS ONE

2. Please provide additional details regarding participant consent. In the ethics statement in the Methods and online submission information, please ensure that you have specified what type you obtained (for instance, written or verbal, and if verbal, how it was documented and witnessed). If your study included minors, state whether you obtained consent from parents or guardians. If the need for consent was waived by the ethics committee, please include this information

“The present research was funded by the University of Rennes 1, Fondation de l’Avenir, the Benjamin Delessert Institute, and INRAE. A. Godet received a PhD grant from the University of Rennes 1. The fNIRS device used in this study was funded by CNRS INS2I and FEDER.”

“The present research was funded by the University of Rennes 1, Fondation de l’Avenir, the Benjamin Delessert Institute, and INRAE. A. Godet received a PhD grant from the University of Rennes 1. The fNIRS device used in this study was funded by CNRS INS2I and FEDER. We also thank Eugenio Valente (Bionic France) for his technical support.”

“The present research was funded by the University of Rennes 1, Fondation de l’Avenir, the Benjamin Delessert Institute, and INRAE. A. Godet received a PhD grant from the University of Rennes 1. The fNIRS device used in this study was funded by CNRS INS2I and FEDER.”

Reviewers' comments:

Reviewer's Responses to Questions

**Comments to the Author**

1. Is the manuscript technically sound, and do the data support the conclusions?

Reviewer #1: Yes

Reviewer #2: Partly

2. Has the statistical analysis been performed appropriately and rigorously? 

Reviewer #1: Yes

Reviewer #2: Yes

3. Have the authors made all data underlying the findings in their manuscript fully available?

Reviewer #1: No

Reviewer #2: Yes

4. Is the manuscript presented in an intelligible fashion and written in standard English?

Reviewer #1: Yes

Reviewer #2: Yes

5. Review Comments to the Author

Reviewer #1: More statistical information needs to be provided. Tables including b, t, p values and effect sizes would be ideal for all analyses.

I was unable to to access the doi link for the raw data set. please ensure this will be accessible for other to be able to run your analyses exactly as reported.

I suggest receiving english language translation editing services for your revised submission.

Reviewer #2: The manuscript addresses an important question in neurofeedback studies in general. Contamination from non-neural sources is a major concern in studies that employ neurofeedback, and therefore, addressing it is necessary before employing feedback. In this regard, the manuscript presents an interesting study on the effects of short-channel signals on EEG recordings; however, there are some concerns that need to be addressed. The authors have used two parameters, short-channels and varying bandpass parameters. There is inadequate information in the introduction and discussion regarding the choice of bandpass parameters and how they are hypothesized to impact signal reduction. The introduction also lacks in providing a clear explanation of the methodology being used. For instance, there is no explanation on whether greater or lower channel activation is desired; Lack of contextual details will limit the utility of this manuscript to a very small subset of neurofeedback scientists.

It appears that the statistical analysis has been performed appropriately and rigorously. The authors used appropriate statistical tests to analyze the data and presented their findings clearly and transparently.

The authors have made all data underlying the findings in their manuscript fully available. They have provided the data as part of the manuscript's supporting information.

The manuscript is presented in an intelligible fashion and written in standard English, however there are some errors; For instance, “The use of this approach in fNIRS-based NF studies remains confidential” on page 10, first paragraph. Why would the use of this methodology be confidential? I would encourage the authors to proofread their manuscript again.

6. PLOS authors have the option to publish the peer review history of their article (what does this mean?). If published, this will include your full peer review and any attached files.

Reviewer #1: **Yes: **Joshua Daniel Upshaw

Reviewer #2: No

---

## [Author Response · Author response to Decision Letter 0]

25 May 2023

PONE-D-22-34461

Subjective feeling of control during fNIRS-based neurofeedback targeting the DL-PFC is related to neural activation determined with short-channel correction

PLOS ONE

Dear Dr. Val-Laillet,

Thank you for submitting your manuscript to PLOS ONE. After careful consideration, we feel that it has merit but does not fully meet PLOS ONE’s publication criteria as it currently stands. Therefore, we invite you to submit a revised version of the manuscript that addresses the points raised during the review process.

We look forward to receiving your revised manuscript.

Kind regards,

Xiong Jiang

Academic Editor

PLOS ONE

ANSWER:

Accordingly, we changed figure’ names following the submission guidelines in the manuscript.

2. Please provide additional details regarding participant consent. In the ethics statement in the Methods and online submission information, please ensure that you have specified what type you obtained (for instance, written or verbal, and if verbal, how it was documented and witnessed). If your study included minors, state whether you obtained consent from parents or guardians. If the need for consent was waived by the ethics committee, please include this information

ANSWER:

We now provide this information in the Materials and Methods section (in the Ethics statement and Participants and study procedure sub-sections).

“The present research was funded by the University of Rennes 1, Fondation de l’Avenir, the Benjamin Delessert Institute, and INRAE. A. Godet received a PhD grant from the University of Rennes 1. The fNIRS device used in this study was funded by CNRS INS2I and FEDER.”

ANSWER:

As requested, this information was stated in the cover letter and added to our funding statement in the online submission form.

“The present research was funded by the University of Rennes 1, Fondation de l’Avenir, the Benjamin Delessert Institute, and INRAE. A. Godet received a PhD grant from the University of Rennes 1. The fNIRS device used in this study was funded by CNRS INS2I and FEDER. We also thank Eugenio Valente (Bionic France) for his technical support.”

“The present research was funded by the University of Rennes 1, Fondation de l’Avenir, the Benjamin Delessert Institute, and INRAE. A. Godet received a PhD grant from the University of Rennes 1. The fNIRS device used in this study was funded by CNRS INS2I and FEDER.”

ANSWER:

As requested, this information was stated in the cover letter and added to our funding statement in the online submission form.

ANSWER:

As suggested, this sentence was removed. Thank you for this reminder.

Reviewers' comments:

Reviewer's Responses to Questions

Comments to the Author

1. Is the manuscript technically sound, and do the data support the conclusions?

Reviewer #1: Yes

Reviewer #2: Partly

2. Has the statistical analysis been performed appropriately and rigorously?

Reviewer #1: Yes

Reviewer #2: Yes

3. Have the authors made all data underlying the findings in their manuscript fully available?

Reviewer #1: No

Reviewer #2: Yes

4. Is the manuscript presented in an intelligible fashion and written in standard English?

Reviewer #1: Yes

Reviewer #2: Yes

5. Review Comments to the Author

Reviewer #1: More statistical information needs to be provided. Tables including b, t, p values and effect sizes would be ideal for all analyses.

I was unable to access the doi link for the raw data set. please ensure this will be accessible for other to be able to run your analyses exactly as reported

I suggest receiving English language translation editing services for your revised submission.

ANSWER: 

We provide answers to all of these questions in the response to specific reviewers’ comments, as detailed below. English language has also been revised in the whole manuscript.

Reviewer #2: The manuscript addresses an important question in neurofeedback studies in general. Contamination from non-neural sources is a major concern in studies that employ neurofeedback, and therefore, addressing it is necessary before employing feedback. In this regard, the manuscript presents an interesting study on the effects of short-channel signals on EEG recordings; however, there are some concerns that need to be addressed. The authors have used two parameters, short-channels and varying bandpass parameters. There is inadequate information in the introduction and discussion regarding the choice of bandpass parameters and how they are hypothesized to impact signal reduction. The introduction also lacks in providing a clear explanation of the methodology being used. For instance, there is no explanation on whether greater or lower channel activation is desired; Lack of contextual details will limit the utility of this manuscript to a very small subset of neurofeedback scientists.

ANSWER: 

Thank you for these important remarks. We have added some information regarding the physical and physiological principles behind the fNIRS technology/method (l. 83-91), as well as the choice of bandpass filter in the introduction (l. 99-104), with a sentence in the last paragraph of the introduction (l. 157-158) stating “since there is no current consensus on filtering option for NF-based NIRS signal analysis”.

For the sake of clarity for the reader, we also added a sentence explaining that increased DL-PFC activity would result in an improvement of cognitive abilities. Line 126 and lines 159-160 

It appears that the statistical analysis has been performed appropriately and rigorously. The authors used appropriate statistical tests to analyze the data and presented their findings clearly and transparently.

The authors have made all data underlying the findings in their manuscript fully available. They have provided the data as part of the manuscript's supporting information.

The manuscript is presented in an intelligible fashion and written in standard English, however there are some errors; For instance, “The use of this approach in fNIRS-based NF studies remains confidential” on page 10, first paragraph. Why would the use of this methodology be confidential? I would encourage the authors to proofread their manuscript again.

ANSWER: 

This sentence was indeed misleading, so we modified it accordingly (l. 117).

6. PLOS authors have the option to publish the peer review history of their article (what does this mean?). If published, this will include your full peer review and any attached files.

Do you want your identity to be public for this peer review? For information about this choice, including consent withdrawal, please see our Privacy Policy.

Reviewer #1: Yes: Joshua Daniel Upshaw

Reviewer #2: No

 

Summary of Article:

One goal of this manuscript was to provide results from a study which examined differences between two separate band-pass filters ranges on the quality of fNIRs measures of hemoglobin based physiological mechanisms underlying brain activity during a 15-trial neurofeedback session. The study further aimed to examine the methodological advantages of including short channel signal as a regressor in their brain activity analysis models with the goal of reducing non-brain related signal during a neurofeedback session. In addition, the study assessed the degree to which individual’s self-reported mental strategies and subjective feelings of control during the neurofeedback session were associated with their brain activity. 

Major Points:

Overall, this manuscript was well written and interesting, and the findings from this study provide valuable insights and information for both researchers and clinicians utilizing neurofeedback and fNIRS neuroscience methodologies. With a few adjustments I think this paper would be a valuable contribution to the field of neuroscience research, specifically for research involving neurofeedback. I think this manuscript would make a strong contribution to the aims of PloS One if the following concerns are addressed. I also want to preface my reviewer comments by saying that I am not an expert in the field of fNIRS or neurofeedback, though I have some experience in this area. I am more well-versed in the cognitive EEG literature. However, I believe if my reviews are addressed, this will help the manuscript be more accessible for a broader audience of readers, which is a goal of PloS One.

A primary concern or area for improvement for this manuscript is a seeming lack of introductory information to set the stage for a broader reader base to better understand the study. 

1. For one, I would like to see more info that describes the rationale for choosing the specific band-pass filters employed in the current study. I would like to see more background information on what filters prior studies have used and what limitations they encountered. 

ANSWER:

As suggested by the reviewer, and based on the complete review of Pinti et al. we have completed the introduction with some details about the rational decision regarding filter options (l. 99-104).

Pinti P, Scholkmann F, Hamilton A, Burgess P, Tachtsidis I. Current Status and Issues Regarding Pre-processing of fNIRS Neuroimaging Data: An Investigation of Diverse Signal Filtering Methods Within a General Linear Model Framework. Front Hum Neurosci [Internet]. 2019 [cited 2019 Dec 13];12. Available from: 

https://www.frontiersin.org/articles/10.3389/fnhum.2018.00505/full

2. Similarly, I would like to see more description of background information on the prior use or non-use, limitations, and strengths of different SC corrections.

ANSWER: 

We now provide a new reference providing detailed information regarding the advantages of SC correction (l. 112-113, 132-133).

Santosa H, Zhai X, Fishburn F, Sparto PJ, Huppert TJ. Quantitative comparison of correction techniques for removing systemic physiological signal in functional near-infrared spectroscopy studies. Neurophotonics. 2020 Jul;7(3):035009. doi: 10.1117/1.NPh.7.3.035009. Epub 2020 Sep 23. PMID: 32995361; PMCID: PMC7511246.

3. In addition, it would be helpful to provide more description of the NF methodologies employed, specifically regarding the choice and purpose of the sources and detectors.

ANSWER: 

We now provide a general description of the purpose of sources and detectors for fNIRS (l. 83-91). Regarding the choice of the channels for DL-PFC neurofeedback, please see below.

Another major concern is the apparent lack of complete statistical values for the analyses. I would like to see beta weights, t-values, p-values, and effect sizes if possible. This is important not only for meta-analyses but also for future work that will utilize this study to inform their research using fNIRS and NF.

ANSWER: 

As suggested by the reviewer, we provided, as linked with the DOI, a table with the statistics for all channels (couples of source-detector and source-SC detector). A sentence was added in the manuscript that indicates the availability of this data in the DOI link (l. 331).

Another major concern I had with reading the methods/results section is not having a clear understanding of which electrodes were utilized in the analyses. Is it possible to provide channel labels or some way to identify specifically which cortical locations were used for analysis?

ANSWER:

Following the reviewers’ suggestion, we now provide information regarding the choice and identification of channels (couple of source-detector) that cover the DL-PFC. The choice of the selected channels covering the DL-PFC is based on Gilman et al. 2022, where they used a layout similar to the one we used here (l. 334-340). A full source / detector / SC detector identification is now provided in Figure 2A and the results and figure legend were modified accordingly.

Gilman JM, Schmitt WA, Potter K, Kendzior B, Pachas GN, Hickey S, Makary M, Huestis MA, Evins AE. Identification of ∆9-tetrahydrocannabinol (THC) impairment using functional brain imaging. Neuropsychopharmacology. 2022 Mar;47(4):944-952. doi: 10.1038/s41386-021-01259-0. Epub 2022 Jan 8. Erratum in: Neuropsychopharmacology. 2022 Feb 28;: PMID: 34999737; PMCID: PMC8882180.

Minor points:

The following points are suggestions for areas for improvement of this paper to improve broader reader comprehension and field specific understanding of the methodological approaches. Keep in mind these are all suggestions as I am not an fNIRS or NF expert. Also, suggestions for various minor grammar/typo/English language issues will be provided as a tracked changes marked-up word doc of the manuscript.

Overall:

1. I suggest sending this manuscript to a writing studio for revision after you make any edits to help with any English language translations issues

ANSWER:

A full careful attention proofreading was made for of the English language revised manuscript was performed. 

2. Consider exploring current APA 7 style heading formatting.

ANSWER: 

We followed the guidelines provided by the editor (APA 7 Style heading formatting).

Paragraph 1 of intro:

1. Run-on sentence. Consider Revising. 

a. “Although the temporal resolution is lower than the resolution achieved by EEG, and only cortical brain activity can be observed, fNIRS appears as to be a good compromise for NF studies and has been increasingly used.”

ANSWER: 

We modified the manuscript accordingly.

Paragraph 2 of intro:

1. Does this mean that the research/clinicians will not share their methodology? If not, what is the reason most NF studies do not use SC corrections? 

a. “The use of this approach in fNIRS-based NF studies remains confidential” 

ANSWER: 

We modified the manuscript accordingly.

Paragraph 4 of intro:

1. Consider providing examples of these strategies.

a. “Moreover, the degree of awareness understanding of the NF training paradigm, the instructions provided (explicit or implicit) and the type of mental strategy used in order to control the region of interest may profoundly affect NF success”

2. In addition, consider describing the differences between mental strategies, in particular, those explored in the current study.

ANSWER: 

We modified the manuscript accordingly.

Paragraph 5 of intro:

1. Consider indicating what the different filter conditions are.

Are you making any specific predictions/hypotheses, if so, considering including them here.

ANSWER: 

We modified the manuscript accordingly by including predictions and hypotheses (l. 156-159). 

Paragraph 2 of methods:

1. Are you basing this sample size from prior work? If so, consider including this info.

ANSWER: 

As explained in the manuscript, and since no similar studies were already performed, a number of 30 participants, as usually in neuroimaging exploratory studies, was considered well suited.

1. Consider including a figure image below this paragraph or in supplementary material if possible. Figure 4 should be labeled Figure 1.

2. Also consider including if more mental effort increased or decreased the position of the gauge.

ANSWER: 

Figure 4 is now Figure 1. Other images names were modified accordingly.

As we did not provide explicit strategy for NF, different strategies were used. We thus cannot demonstrate whether mental effort did increase or decrease the position of the gauge. 

Paragraph 4 of methods:

1. Is it possible to rotate figure 1A to the right and down, and include an XYZ axis? Not necessary, per se, but could be helpful for comprehension. Also, it could be useful to label your electrodes according to the 10-20 system or another channel naming convention.

2. Consider briefly explaining what increases and decreases in oxy-Hb mean in terms of cognitive function.

ANSWER: 

We now only provide identification of sources, detectors and SC detectors in Figure 2A., since it is not common to add EEF localizers in NIRS analysis. We also wanted not to overload this figure. 

We now briefly explain the link between oxy-Hb and brain activity (l. 81-91).

Paragraph 5 of methods:

1. Consider moving this to earlier in the manuscript. Perhaps at the end of the participants and procedure section. 

a. “See figure 4 for an overview of the protocol design and scheme of NF sessions.”

2. Again I think Figure 4 should be mentioned sooner and thus changed to Figure 1.

ANSWER: 

We thanks the reviewer for this suggestion and modified the manuscript accordingly.

Paragraph 6 of methods:

1. Could you provide information about the specifics of the computer monitor? The model, resolution, refresh rate, the distance participants sat from the monitor? If someone were to try and replicate this study to get the same results, what exactly would they need to do to recreate the protocol conditions?

ANSWER: 

The participants sat at a distance of 80 cm from the monitor (HP Compaq LA2045wg, 1680x1050, 59.94 Hz). We also added information regarding the monitor line (l. 249-250).

Paragraph 7 of methods:

1. Can you provide an example? What is a to-low filter range? Same for too-high filter?

a. “a too-low low-pass filter might remove too much relevant neural signal, but a too- high low-pass filter”

ANSWER: 

These questions were answered above (point 1.).

2. This is a complicated model for broader audiences to grasp. It could help to include a figure that depicts your models used. You could then refer to that figure in future sections.

ANSWER: 

Although GLM models are common tools for statistical analysis, we do understand that the statistical model might not be intuitive.We added a reference explaining the model used (Barker et al. 2013).

Barker JW, Aarabi A, Huppert TJ. Autoregressive model based algorithm for correcting motion and serially correlated errors in fNIRS. Biomed Opt Express. 2013;4(8):1366. 

3. I’m not familiar with this type of filter. Is it commonly utilized in the NF field? 

If not, consider providing more information.

a. “via pre-whitening filters”

ANSWER: 

The pre-whitening filter is commonly used in fNIRS signal analysis, as slow physiological oscillations are sampled rapidly causing nearby time points in the fNIRS signal to be highly correlated, leading to serially correlated error terms in the GLM. Pre-whitening is used to detrend, and make the measurement "White", namely independent between each measurement.

Paragraph 8 of methods:

1. When you say “around”, was there a range? Is it important to clarify this?

a. “was around 30 mm for a long channel and 8 mm for a short-channel”

ANSWER: 

Indeed, this distance varies between 30 to 42 mm according to the commercial layout. We consequently modified this sentence in the manuscript.

2. Consider elaborating on this point in the intro as it seems to be a primary goal of your project.

ANSWER: Indeed, the distance for short-channels classically ranges between 0.5 and 1.5 cm [42], but a distance under 10 mm is recommended [43]. A sentence was added in the introduction (l. 132-133).

3. Is it possible to provide the electrode channel labels according to the 10-20 system? E.g. Fz Cz F3

ANSWER: 

We apologized for the lack of clarity. The source and detector localization were placed with regards to EEG layout. For the sake of clarity, and as commonly described for fNIRS study, we modified the Figure 2A by adding source and detector names.

S1 F3

S2 C3

S3 F3

S4 Fz

S5 Fpz

S6 AF4

S7 F4

S8 AF8

D1 F5

D2 F1

D3 C1

D4 AFz

D5 F2

D6 Fp2

D7 F6

DL-PFC NF channels :

- S1D1 : F3 – F5

- S1D2 : F3 – F1

- S7D5 : F4 – F2

- S7D7 : F4 – F6

a. “Three regions of interest”

4. This was difficult to read and comprehend. Is it possible to rewrite this? Also, is this a common practice for NF research? Can you cite exemplars?

a. “Given that within a single NF session, each participant was possibly exploring different strategies, not all NF trials were expected to yield successful NF control. We thus decided to investigate the individual channel response in relation with the entire 15 trials, but also the 10 and 5 best trials, determined as the best averaged t-value in the four NF channels computed for each trial.”

ANSWER: 

We modified the manuscript accordingly. Line 285-286

Paragraph 8 of methods:

1. Can you indicate which figure numbers here. You are referencing?

a. “detailed in the figures’ legends”

2. Which figure? Include figure number?

a. “the figures’ details.” 

ANSWER: 

We modified the manuscript accordingly

Subheading 1 of results:

1. Consider replacing with a broader, more general results heading titles.

a. “One single DL-PFC-targeted NF session promotes effective DL- 

b. PFC activation for at least 10 trials within 15.

ANSWER: The heading title was modified accordingly.

Paragraph 1 of results:

1. Do you have tables to reference with the statistical outputs? The B values, t-values, p-values, effect sizes? This would be useful for multiple reasons, especially for meta-analyses.

2. Did you elaborate your predictions in the intro? If not, it would be helpful to do so for the reader to understand what you were measuring.

a. “we expected an increase in HbO2 and a decrease in HbR”

ANSWER: A table with all results is now provided as a DOI.

It is commonly accepted that an activated channel shows an increase in HbO2 and a decrease in HbR.

Paragraph 2 of results:

1. Not sure which channels these are. Consider providing channel labels on figures and in-text.

a. “four NF channels”

2. Here would be a good place to include the findings for including short-channel correction models similar to above.

a. “without SC correction”

ANSWER: 

We now identify the “four channels” in the text.

Subheading 2 of results:

1. Consider replacing with more general results section title.

a. “Depending on the BP filtering characteristic, the activation within the NF channels is moderately impacted by physiological noise and is not associated with other frontal and motor brain regions activation.”

ANSWER: The heading title was modified accordingly

Paragraph 7 of results:

1. Both this paragraph and the one prior could benefit comprehension if there was an accompanying table that indicated the results. For me, the figures are not intuitive to navigate. Also, that table could include regression beta’s, t’s, and p values. Effects sizes would also be great.

a. “For the BP 0.01-0.2 Hz filter”

ANSWER: A table with all channels data is now provided with a DOI.

Paragraph 8 of results:

1. It is not clear to me at this point why the narrower band pass filter is better. Consider providing a brief explanation of why this is prior to this sentence.

ANSWER: We now provide an additional sentence explaining the advantages of the narrower filter. Lines 385-387.

Paragraph 9 of results:

1. I don’t quite understand what you are comparing for significance here. Is it the number of participants with activation in at least one source/detector value compare participants with 0? Why are you comparing these? Are you trying to demonstrate that when SC are regressed out when using the .01-.20 BP filter that there are significantly more participants showing activation? Does that contradict the statement in the earlier paragraph that says the .01-.09 BP filter is better?

a. “Chi-square test for the proportion of the participants with activated channel in function of DL-PFC+SC activation or not, * p<0.05.”

ANSWER: We indeed compared the number of participants who showed at least one activated channel. .01-.20 BP filter yielded a higher number of participants with at least one activated channel, but this was associated with activation in other frontal and motor brain regions. Inversely, the DL-PFC NF targets an increased activation that was restricted to the DL-PFC. This global brain activation .01-.20 BP filter suggests that the larger bandpass filter might not yield a specific outcome of brain activation.

Paragraph 10 of results:

1. I’m not sure if this way of phrasing that you are regressing out SC’s is intuitive. Consider stating “for SC regressed models”.

a. “DL-PFC+SC”

2. All p-values should be reported. Consider making a table for this result section similar to the ones above.

3. Why is this data not shown? Consider including in a table.

a. “(data not shown).”

ANSWER: 

The sentence was modified accordingly.

A table is now provided in the DOI link.

All sentences with “data not shown” were removed.

Paragraph 11 of results:

1. This section could be elaborated upon with more detail.

ANSWER: We provided more details in this paragraph. 

Paragraph 1 of discussion:

2. After this sentence, you should provide a brief description of the study aims and procedures.

a. “In the present study, 30 individuals performed a single session of fNIRS-based neurofeedback (NF) targeting the DL-PFC bilaterally that consisted in of 15 trials of NF alternating with 30-sec alternating resting periods.”

ANSWER: Accordingly, we provided a brief discussion of the study aims and procedure.

---

## [Decision Letter · Decision Letter 1]

17 Jul 2023

PONE-D-22-34461R1Subjective feeling of control during fNIRS-based neurofeedback targeting the DL-PFC is related to neural activation determined with short-channel correctionPLOS ONE

Dear Dr. Val-Laillet,

Thank you for submitting your manuscript to PLOS ONE. After careful consideration, we feel that it has merit but does not fully meet PLOS ONE’s publication criteria as it currently stands. Therefore, we invite you to submit a revised version of the manuscript that addresses the points raised during the review process.

We look forward to receiving your revised manuscript.

Kind regards,

Xiong Jiang

Academic Editor

PLOS ONE

Journal Requirements:

Reviewers' comments:

Reviewer's Responses to Questions

**Comments to the Author**

1. If the authors have adequately addressed your comments raised in a previous round of review and you feel that this manuscript is now acceptable for publication, you may indicate that here to bypass the “Comments to the Author” section, enter your conflict of interest statement in the “Confidential to Editor” section, and submit your "Accept" recommendation.

Reviewer #1: All comments have been addressed

2. Is the manuscript technically sound, and do the data support the conclusions?

Reviewer #1: Yes

3. Has the statistical analysis been performed appropriately and rigorously? 

Reviewer #1: Yes

4. Have the authors made all data underlying the findings in their manuscript fully available?

Reviewer #1: Yes

5. Is the manuscript presented in an intelligible fashion and written in standard English?

Reviewer #1: Yes

6. Review Comments to the Author

Reviewer #1: Great work on this revision! I appreciate your time and energy in making this paper more widely understood to less expert audiences and for improving the level of technical detail and specificity for replication purposes. There are a few very minor grammar and typo edits to be made and I think this manuscript will be ready for publication depending on the what the editors believes is best.

7. PLOS authors have the option to publish the peer review history of their article (what does this mean?). If published, this will include your full peer review and any attached files.

Reviewer #1: **Yes: **Joshua D. Upshaw

---

## [Author Response · Author response to Decision Letter 1]

18 Jul 2023

Reviewer #1: Great work on this revision! I appreciate your time and energy in making this paper more widely understood to less expert audiences and for improving the level of technical detail and specificity for replication purposes. There are a few very minor grammar and typo edits to be made and I think this manuscript will be ready for publication depending on the what the editors believes is best.

Answer: We would like to thank you for pointing out persistent grammatical errors, which have been corrected according to your suggestions. We have also added the year of reference 18 (Pinti, 2019) and corrected a few errors in the citation style of references.

---

## [Decision Letter · Decision Letter 2]

1 Aug 2023

Subjective feeling of control during fNIRS-based neurofeedback targeting the DL-PFC is related to neural activation determined with short-channel correction

PONE-D-22-34461R2

Dear Dr. Val-Laillet,

We’re pleased to inform you that your manuscript has been judged scientifically suitable for publication and will be formally accepted for publication once it meets all outstanding technical requirements.

Kind regards,

Xiong Jiang

Academic Editor

PLOS ONE

Additional Editor Comments (optional):

Reviewers' comments:

Reviewer's Responses to Questions

**Comments to the Author**

1. If the authors have adequately addressed your comments raised in a previous round of review and you feel that this manuscript is now acceptable for publication, you may indicate that here to bypass the “Comments to the Author” section, enter your conflict of interest statement in the “Confidential to Editor” section, and submit your "Accept" recommendation.

Reviewer #1: All comments have been addressed

2. Is the manuscript technically sound, and do the data support the conclusions?

Reviewer #1: Yes

3. Has the statistical analysis been performed appropriately and rigorously? 

Reviewer #1: Yes

4. Have the authors made all data underlying the findings in their manuscript fully available?

Reviewer #1: Yes

5. Is the manuscript presented in an intelligible fashion and written in standard English?

Reviewer #1: Yes

6. Review Comments to the Author

Reviewer #1: Well done! I appreciate your hard work on this paper. I think this manuscript will offer neurofeedback users helpful methodological information to use in their research.

7. PLOS authors have the option to publish the peer review history of their article (what does this mean?). If published, this will include your full peer review and any attached files.

Reviewer #1: **Yes: **Joshua D. Upshaw

---

## [Editor Report · Acceptance letter]

7 Aug 2023

PONE-D-22-34461R2 

Subjective feeling of control during fNIRS-based neurofeedback targeting the DL-PFC is related to neural activation determined with short-channel correction 

Dear Dr. Val-Laillet:

I'm pleased to inform you that your manuscript has been deemed suitable for publication in PLOS ONE. Congratulations! Your manuscript is now with our production department. 

Kind regards, 

on behalf of

Dr. Xiong Jiang 

Academic Editor

PLOS ONE